**RESEARCH**

# Host-microbiome protein-protein interactions capture disease-relevant pathways

Hao Zhou[1†], Juan Felipe Beltrán[2†] and Ilana Lauren Brito[2*]

*Correspondence:
ibrito@cornell.edu
†Hao Zhou and Juan Felipe
Beltrán contributed equally
to this work.
[2] Meinig School
of Biomedical Engineering,
Cornell University, Ithaca,
NY, USA
Full list of author information
is available at the end of the
article

## Abstract

**Background:** Host-microbe interactions are crucial for normal physiological and immune system development and are implicated in a variety of diseases, including inflammatory bowel disease (IBD), colorectal cancer (CRC), obesity, and type 2 diabetes (T2D). Despite large-scale case-control studies aimed at identifying microbial taxa or genes involved in pathogeneses, the mechanisms linking them to disease have thus far remained elusive.

**Results:** To identify potential pathways through which human-associated bacteria impact host health, we leverage publicly-available interspecies protein-protein interaction (PPI) data to find clusters of microbiome-derived proteins with high sequence identity to known human-protein interactors. We observe differential targeting of putative human-interacting bacterial genes in nine independent metagenomic studies, finding evidence that the microbiome broadly targets human proteins involved in immune, oncogenic, apoptotic, and endocrine signaling pathways in relation to IBD, CRC, obesity, and T2D diagnoses.

**Conclusions:** This host-centric analysis provides a mechanistic hypothesis-generating platform and extensively adds human functional annotation to commensal bacterial proteins.

**Keywords:** Gut microbiome, Protein-protein interactions, Metagenomics, Human disease

## Introduction

Metagenomic case-control studies of the human gut microbiome have implicated bacterial genes in a myriad of diseases. Yet, the sheer diversity of genes within the microbiome [1] and the limitations of functional annotations [2] have thwarted efforts to identify mechanisms linking bacterial genes with host health. In the cases where functional annotations exist, they tend to reflect molecular functions often relevant to cellular survival under cultured conditions (e.g. DNA binding, post-translational modification). Few annotations reflect microbial proteins' roles as they related to host cell signaling and

homeostasis. Associating any commensal bacterial gene and a host pathway has thus far required experimental approaches catered to each gene or gene function [3, 4].

For pathogens, protein-protein interactions (PPIs) involving human proteins, obtained through in-depth structural studies of individual proteins [4–6], as well as large-scale whole-organism interaction screens [7–17], provide direct links between microbial proteins and host pathways. Several canonical protein-mediated microbe-associated molecular patterns (MAMPs) that directly trigger host-signaling pathways through pattern recognition receptors present on epithelial and immune tissues [18] are conserved between pathogens and commensals [19], such as that between flagellin with Toll-like receptor 5 (TLR5). We hypothesized that host-microbiome PPIs may be more widespread and that they may underlie microbiome-mediated disorders. A handful of studied interactions suggest that this may be the case (Additional file 1: Table S1). For example, the *Akkermansia muciniphila* protein P9 binds intercellular adhesion molecule 2 (ICAM2) to increase thermogenesis and glucagon-like peptide-1 (GLP-1) secretion, a therapeutic target for type 2 diabetes (T2D) [20]; the protein Fap2 from *Fusobacterium nucleatum* binds T cell immunoreceptor with Ig and ITIM domains (TIGIT), inhibiting natural killer cytotoxicity; and ubiquitin mimics encoded by both pathogens [21] and gut commensals [22] play a role in modulating membrane trafficking.

Seeking to examine host-microbiome interactions more broadly, we extracted all experimentally verified inter-species PPIs involving human proteins from large-scale interaction databases (IntAct [23], BioGRID [24], HPIdb [25]) and a set of manually curated publications. We found 15,252 unique inter-species PPIs, yet only a handful of these involve proteins pulled from the human gut microbiome (Additional file 2: Fig. S1). In the absence of structure and direct experimental data, sequence identity methods have been used to infer host-pathogen PPI networks for single pathogens [26–28], but such approaches have not yet been applied at the community level, as would be required for the human gut microbiome. Here, we explore proteins within the gut microbiome that have homology to interactors that bind human proteins. To hone in on only those interactions relevant to disease, we examine their abundances across nine metagenomic cohorts, ascribing putative disease-relevant roles for hundreds of microbiome proteins.

## Results

### Mapping microbiome proteins to known PPIs identifies potential mechanistic links to disease

To distinguish PPIs that may be associated with health versus disease, we compared community-level PPI profiles in large case-control cohorts of well-established microbiome-associated disorders—namely inflammatory bowel disease (IBD) [29, 30], colorectal cancer (CRC) [31–34], obesity [35], and T2D [36, 37] (Fig. 1A, Additional file 3: Table S2). In order to build community-level PPI profiles, we associated gene family abundances in these nine studies to a newly constructed database of bacterial human-protein interactors and the bacterial members of their associated UniRef clusters [38] (Additional file 2: Fig. S2). For further assurance, we required microbiome proteins to have high amino-acid similarity (at least 70%) with the specific proteins with experimental evidence of interacting with human proteins. We found that interspecies bacterial-human protein interface residues, in general, are highly similar, or even identical,

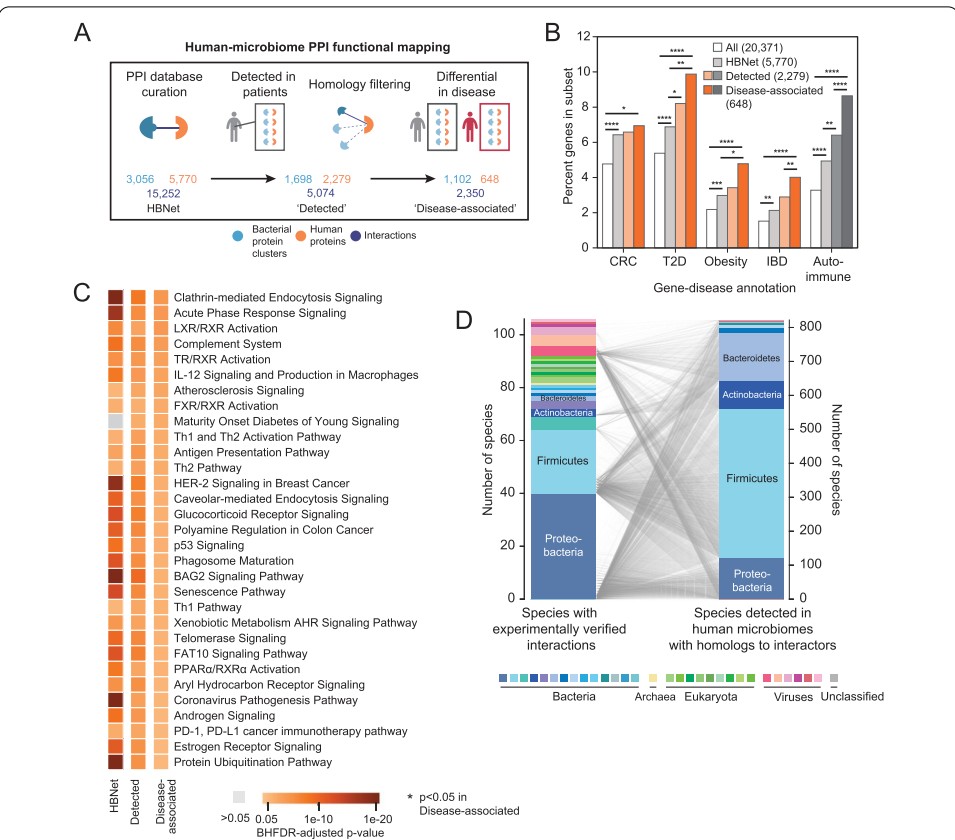

**Fig. 1** Human proteins differentially targeted by the microbiome in disease are enriched for relevant gene-disease associations. **A** The number of interspecies bacterial protein clusters (blue), human proteins (orange), and interactions (dark blue) in the human-bacteria PPI network; the number of bacterial protein clusters detected in patients from nine metagenomic studies that also have homology to experimentally verified interactors and their putative human interactors; and the number of bacterial clusters and human proteins associated with disease through our metagenomic machine learning approach, by comparing abundances in cases (gray) and control (red). **B** Proportions of human proteins implicated in disease, according to their GDAs (GDAs > 0.1) in DisGeNET, within: all reviewed human proteins; HBNet; human interactors with detected bacterial proteins; and those human interactors with feature importances above the 90th percentile in their respective cohorts. *p* values for enrichments are depicted by: * *p*<0.05; ** *p*<0.01; *** *p*<10$^{-3}$; **** *p*<10$^{-4}$ (chi-square test). Total numbers of each set are noted in the legend. **C** Human cellular pathways (annotated by IPA) enriched in the set of human proteins within HBNet (left) and those detected across all nine metagenomic case-control studies (right) colored according to their Benjamini-Hochberg false discovery rate (BHFDR)-adjusted *p* value. Only those pathways with BHFDR-adjusted < 0.05 in the disease-associated sets are shown. *p* values for enrichments are depicted by: * *p*<0.05; ** *p*<0.01; *** *p*<10$^{-3}$; **** *p*<10$^{-4}$ (Fisher's exact test). **D** 106 species (left) with experimentally verified proteins in 3056 bacterial protein clusters are mapped to 821 bacterial species (right) with homologs detected in patients' metagenomes (right), representing a total of 1698 clusters. Species are colored according to phylum

between members of the same UniRef cluster filtered in the same manner (Additional file 2: Fig. S3). We noticed that this process filtered out proteins exclusively in pathogenic organisms, such as the *Clostridium difficile* toxin B (TcdB) which binds frizzled 2 (FZD2), or expressed predominantly by pathogenic isolates, such as *Finegoldia magna* protein L, which binds immunoglobulin L chains [39].

After applying a random forest classifier trained on each disease cohort (Additional file 2: Fig. S6), we find 1,102 commensal bacterial protein clusters associated with disease, by virtue of their putative interactions with 648 human proteins (Additional

file 4: Table S3). Focusing on putative microbiome interactors with strong associations with disease weeds out a greater percentage of interactions initially detected by yeast-2-hybrid (Y2H) methods and enriches for those that are based on affinity techniques (Additional file 2: Fig. S4), and consequently removes the most "sticky" bacterial proteins (Additional file 2: Fig. S5). The human protein with the highest degree remaining is nuclear factor NF-κB p105 subunit (NFKB1), a protein involved in immunodeficiency and bacterial infection, which was differentially targeted in CRC (in Vogtmann et al.).

To test whether the human-microbiome PPIs identified in each disease cohort were relevant to that disease, we analyzed the roles of the human protein targets compared with the entirety of the human proteome, as well as those that were detected, but not associated with disease. We found that moving from the 5770 human proteins within the interaction network ("HBNet"), to the 2279 human proteins with bacterial interactors detected in human microbiomes ("Detected"), to the 648 that are associated with disease ("Disease-associated"), we observe increasing enrichment for proteins with previously-reported gene-disease associations (GDA) in CRC, diabetes, obesity, and IBD (Fig. 1B). Disease-specific enrichments are even more pronounced when examining each specific disease cohort (Additional file 2: Fig. S7). However, we see enrichment for all four microbiome-associated disorders in each of the cohorts, reflecting their associated relative risks [40–44]. Out of all of the proteins with any GDA in the disease-associated set, 45.2% have more than one GDA for our diseases of interest. We suspected this may extend to autoimmune diseases, which are increasingly studied in the context of the gut microbiome [45]. As expected, we found enrichment of GDAs for autoimmune disorders in the human proteins implicated by our method as well (Fig. 1B, Additional file 2: Fig. S7). Additionally, we associated over half of the PPIs that have been well-studied for both binding and their effect on human cellular physiology or disease pathophysiologies (Additional file 1: Table S1) with one or more metagenomic studies. The concordance between known gene-disease annotation and disease association of each microbiome cohort demonstrates the utility of using PPIs to capture molecular heterogeneity that underlies microbiome-related disease.

In evaluating the statistical significance of recurrent human functional annotations, we performed pathway enrichment analysis on the implicated human proteins and find proteins with established roles in cellular pathways coherent with the pathophysiology of IBD, CRC, obesity, and T2D (Fig. 1C), namely those involving the immune system, apoptosis, oncogenesis, and endocrine signaling pathways. Enriched pathways were not specific to disease, but included human proteins across the four types of disease cohorts analyzed. Disease-associated human targets were enriched in pathways including those involved in bacterial pathogenesis and underlying inflammation, such as the IL-12 signaling pathway and clathrin-mediated endocytosis signaling. These pathways were expected due to shared evolutionary histories between the screened pathogens and gut microbiota and opportunism within the microbiome. Enriched pathways also included bile salt metabolism and cholesterol metabolism (LXR/RXR, TX/RXR, and FXR/RXR activation pathways), which are also tied to immune evasion [46, 47], expanding the role of the microbiota in these pathways beyond their enzymatic functions.

Within these pathways, we see specific examples of known molecular mechanisms for these diseases now implicated with microbiome-host PPIs: Actin-related protein

2/3 complex subunit 2 (ARPc2) (associated in both IBD and CRC cohorts) regulates the remodeling of epithelial adherens junctions, a common pathway disrupted in IBD [48]. We see the targeting of mitogen-activated protein kinase kinase kinase kinase 1 (MAP 4K1) enriched in the Zeller et al. CRC cohort, which is in line with its role in inflammation [49]. DNA methyltransferase 3a (DNMT3A) is involved in chromatin remodeling and has been shown to be important for intestinal tumorigenesis [50], serve as a risk loci in genome-wide association studies (GWAS) studies for Crohn's disease [51], mediates insulin resistance [52] and has aberrant expression in adipose tissue in mice [53]. Concordantly, it was associated with the CRC, IBD, T2D, and obesity microbiome studies we examined. This host-centric annotation is useful beyond the large-scale analysis of metagenomic data, as it broadly enables hypothesis-driven research into the protein-mediated mechanisms underlying microbiome impacts on host health.

### Microbiome proteins access human proteins by various means

Although the set of experimentally verified interactions (HBNet) includes interactions originating from 82 unique bacterial species, an initial concern was the disproportionate number of bacteria-human PPIs derived from high-throughput screens performed on a smaller number of intracellular pathogens, e.g., *Salmonella enterica* [9], *Yersinia pestis* [8, 10], *Francisella tularensis* [8], *Acinetobacter baumannii* [14], *Mycobacterium tuberculosis* [11], *Coxiella burnetii* [12], *Chlamydia trachomatis* [13] and *Legionella pneumophila* [15], *Burkholderia mallei* [16], and *Bacillus anthracis* [8], as well as one extracellular pathogen *Streptococcus pyogenes* [17] (Additional file 5: Table S4). Despite this bias, we find that homologs detected in patient microbiomes come from a set of 821 species that better reflects the phyla typically associated with human gut microbiomes (Fig. 1D).

We next examined the localization of human protein targets. We see enrichment of genes expressed in epithelium, liver, adipose tissue, and blood components in the "Detected" and "Disease-associated" human targets (Fig. 2A). Although we presume many of the interactions occur within in the epithelial layer of the gastrointestinal tract, disease-associated human interactors were not especially localized to gastrointestinal tissue, nor any tissue in particular, with the exception of bone marrow ($p$=0.047, chi-square test) (Additional file 2: Fig. S8). Impaired intestinal barrier function and the translocation of commensal bacteria, both of which feature in the pathogenesis of IBD [54], CRC [55], and other microbiome-associated disorders [56], allow bacterial proteins to access tissues exterior to the gut. Nevertheless, we suspect that the absence of enrichment in gut tissues largely reflects the human tissues, cells, and fluids used for experimental interaction screening (e.g., HeLa cells [9], HEK293T [13], macrophages [9], plasma [17], saliva [17], spleen [8, 10], and lung [14]), thereby selecting proteins with more general expression patterns. This data underscores the need for screening using gastroenterological protein libraries to identify gut-specific host-microbiome PPIs.

At the cellular level, microbial proteins can access human proteins via several well-established means (Fig. 2B). Canonical MAMPs tend to involve surface receptors (e.g., TLRs, Nod-like receptors), comprising 59.2% of the disease-associated interactors (Fig. 2C), although we cannot confirm their orientation. We expect that this may be an underestimate of the interactions involving human membrane interactors, as solubility

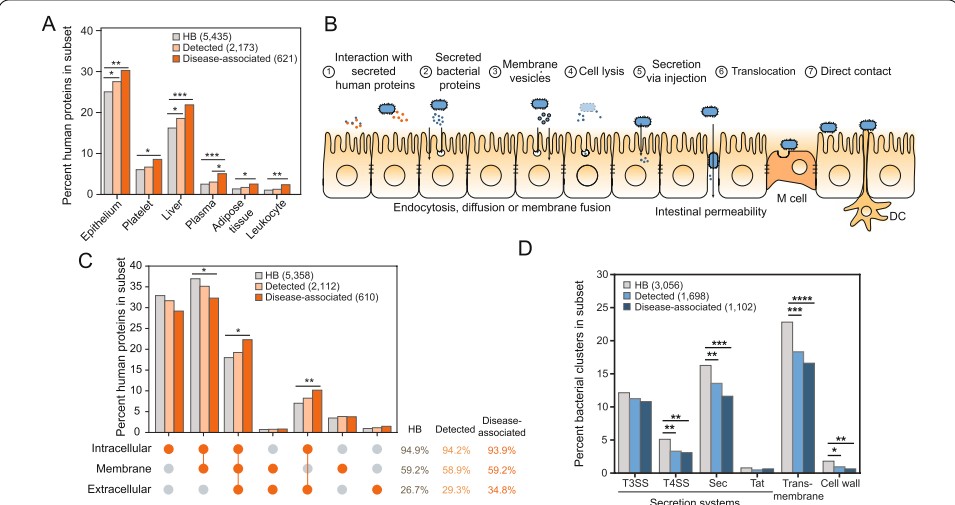

**Fig. 2** Bacterial proteins gain access to human proteins through a variety of mechanisms. **A** Proportions of human proteins in the HBNet, Detected and Disease-associated subsets are plotted according to their enrichments in tissues and fluids, as annotated using DAVID. Only those with significant enrichment between any two subsets are shown. *p* values for enrichments are depicted by * *p*<0.05; ** *p*<0.01; *** *p*<0.001; **** *p*<0.0001 (EASE Score provided by DAVID, a modified Fisher's exact *P* value; FDR-adjusted). Total numbers of each set are noted in the legend. **B** A schematic depicting potential opportunities for bacterial proteins to access human proteins. Interactions may involve (1) secreted human proteins, (2) bacterial proteins secreted into the extracellular space; (3) membrane vesicles that are endocytosed or can fuse with human cell membranes; (4) bacterial cellular lysate; (5) proteins injected into human cells by T3SS, T4SS, and T6SS, (6) cells and their products that translocate as a result of barrier dysfunction or "leaky gut", and/or (7) direct contact with M cells, dendritic cells (DC), or epithelial cells. **C** Proportions of human proteins in the HBNet, Detected and Disease-associated subsets, are plotted according to their subcellular locations, as annotated using Gene Ontology Cellular Component, is depicted. *p* values for enrichments are depicted by: * *p*<0.05; ** *p*<0.01; *** *p*<0.001; **** *p*<0.0001 (chi-square test). Total percentages for these subsets are listed at right, along with p-values. Total numbers of each set are noted in the legend. **D** Proportions of bacterial gene clusters in the HBNet, Detected and Disease-associated subsets are plotted according to their transmembrane and secretion predictions, annotated using TMHMM, EffectiveDB, and SignalP. *p* values for enrichments are depicted by * *p*<0.05; ** *p*<0.01; *** *p*<0.001; **** *p*<0.0001 (chi-square test). Total numbers of each set are noted in the legend

issues preclude their representation in interaction screens. In addition to canonical MAMP receptors, newly described surface receptors include adhesion G protein-coupled receptor E1 (ADGRE1), a protein involved in regulatory T cell development [57]; and receptor-type tyrosine-protein phosphatase mu (PTPRM), involved in cadherin-related cell adhesion [58], among others. Alternatively, interactions may involve human proteins that are secreted, as evidenced by several established host-microbiome PPIs (Additional file 1: Table S1), including the extracellular matrix protein laminin [59] and immune modulators, such as extra-cellular histones [60, 61]. Secreted proteins make up 34.8% of the disease-targeted human interactors, and include these, in addition to the cytokine IL-8, galectin-3, and complement 4A.

Interestingly, a large number of disease-associated human interactors (178 proteins or 29.1%) are exclusively intracellular (Fig. 2C), suggesting additional interaction schemes. MAM (microbial anti-inflammatory molecule), a secreted protein from *Faecalibacterium prausnitzii*, can inhibit NF-κB signaling and increase tight junction integrity, whether it is introduced via gavage in mouse models, or when it is ectopically expressed from within intestinal epithelial cells in vitro [62], suggesting that it is uptaken by cells

in vivo. Bacterial products or, in some cases, intact bacteria, may be endo-, pino-, or transcytosed, a process that can be initiated by receptors [63, 64], allowing bacterial proteins to access cytoplasmic and even nuclear targets. Alternatively, membrane vesicles, decorated with proteins and carrying periplasmic, cytoplasmic, and intracellular membrane proteins as cargo, can be uptaken by human cells via endocytosis or membrane diffusion [65]. Although membrane vesicles have been well-documented in Gram-negative bacteria, an example of vesicle production by Gram-positive segmented filamentous bacteria was recently shown to interact with intestinal epithelial cells and promote the induction of Th17 cells [66].

Accordingly, bacterial proteins interacting with human secreted and surface proteins would be expected to contain signatures of surface localization or extracellular secretion. Indeed, we find that 12.2% of the disease-associated microbiome proteins are predicted to contain signal peptides allowing for secretion by the Sec or Tat pathways (Fig. 2D), which are ubiquitous across phyla (Additional file 2: Fig. S9). These systems typically work alongside additional secretion systems to situate proteins in the cell membrane or secrete them extracellularly, though their associated signal peptides are more difficult to predict [67, 68]. Another 16.6% of disease-associated microbiome proteins are predicted to be transmembrane, albeit with unknown orientation. Surface localization would potentially allow for direct contact between human proteins and either live or intact bacteria, or proteins on the surface of bacterially-produced membrane vesicles. A small number of proteins were found destined for the cell wall (Fig. 2D). To our surprise, secreted and surface proteins were found to be negatively enriched in the disease-associated bacterial interactors.

Finally, type 3, type 4, and type 6 secretion systems (T3SS, T4SS, and T6SS) can be used to secrete proteins directly into human cells. Proteins with T3SS and T4SS signals make up a significant (13.6%), albeit a diminishing portion of the disease-associated microbiome proteins (Fig. 2D). These proteins are mostly derived from gut Proteobacteria, to which these systems are generally restricted [69] (Fig.2D, Additional file 2: Fig. S9). Based on the bacterial cluster representatives in the microbiomes from these nine cohorts, we find evidence that at least 79.0% (94/119) and 58.9% (20/34) of disease-associated clusters predicted to be secreted by T3SS and T4SS, respectively, have representative proteins found in organisms with the corresponding secretion systems (T6SS were excluded due to the limited availability of prediction tools). Nevertheless, the extent to which these systems, and orthologous systems in Gram-positive bacteria [70], play a role in host-microbiome protein trafficking remains unknown. In total, this data suggests that there is not one single mechanism dominating host-microbiome interactions, but that interactions are facilitated by several means.

## Discussion

### Microbiome proteins gain host-relevant "moonlighting" annotations

One of the major advantages of our work is that through this new interaction network, we vastly improve our ability to annotate host-relevant microbiome functions. 13.5% of our disease-associated bacterial clusters contain no members with annotated microbial pathways/functions in KEGG (Kyoto Encyclopedia of Genes and Genomes) [71] (Fig. 3A). Using similar homology searching against bacterial interactors, most of

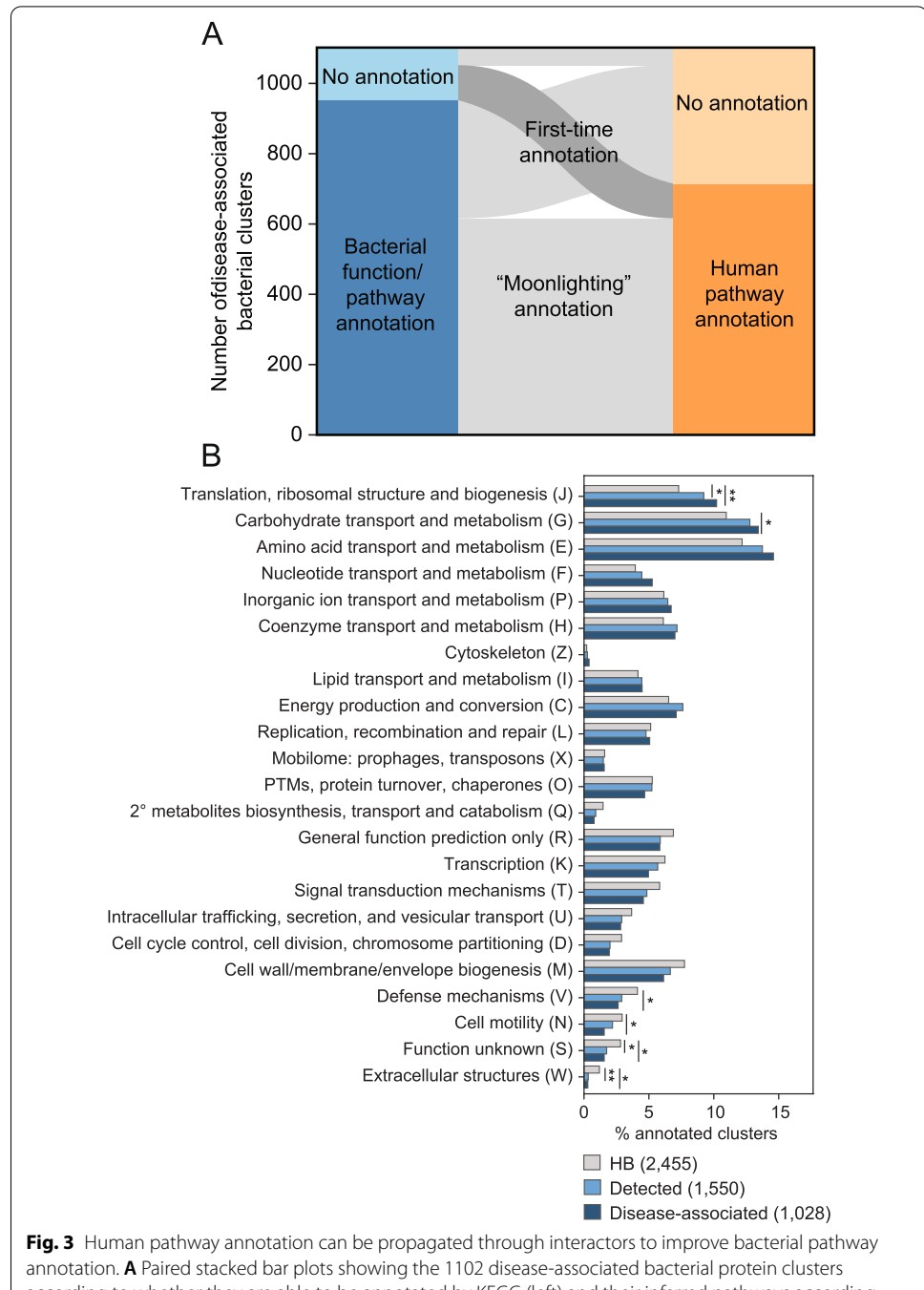

**Fig. 3** Human pathway annotation can be propagated through interactors to improve bacterial pathway annotation. **A** Paired stacked bar plots showing the 1102 disease-associated bacterial protein clusters according to whether they are able to be annotated by KEGG (left) and their inferred pathways according to the human proteins they target (right), as annotated by WikiPathways [77]. **B** Proportions of the bacterial clusters in the HBNet, Detected and Disease-associated subsets according to their COG functional categories are plotted. *p* values are depicted by * *p*<0.05; ** *p*<0.01; *** *p*<0.001; **** *p*<0.0001 (chi-square test). Total numbers of each set are noted in the legend

these genes can now be annotated according to the pathways of their human targets, obtaining a putative disease-relevant molecular mechanism (Additional file 2: Fig. S10). Interestingly, most of the bacterial clusters with KEGG pathway annotations also gain a secondary human pathway annotation. Of those that could be annotated,

disease-associated clusters are involved primarily in translation and central metabolism (Fig. 3B). This dual function is not entirely surprising, as a number of these have orthologs that have been previously identified as bacterial 'moonlighting' proteins, which perform secondary functions in addition to their primary role in the cell [72]. *Mycoplasma pneumoniae* GroEL and *Streptococcus suis* enolase, a protein involved in glycolysis, bind to both human plasminogen and extra-cellular matrix components [73, 74]. *Mycobacterium tuberculosis* DnaK signals to leukocytes causing the release of the chemokines CCL3-5 [75]. *Streptococcus pyogenes* glyceraldehyde-3-phosphate dehydrogenase (GAPDH), canonically involved in glycolysis, can be shuffled to the cell surface where it plays a role as an adhesin, and can also contribute to human cellular apoptosis [76]. These examples distinctly illustrate how bacterial housekeeping proteins are used by pathogens to modulate human health. In this study, we uncover commensal proteins that similarly may have "interspecies moonlighting" functions and appear to be pervasive throughout our indigenous microbiota. Although this method generates provocative hypotheses that directly link human-associated microbiota to disease, functional experiments and structural evidence are greatly needed to confirm these specific host-microbiome PPIs.

### Microbiome proteins may act on human targets as therapeutic drugs

We find that many disease-associated human proteins are known drug targets (Additional file 6: Table S5). For example, nafamostat mesylate is an anticoagulant that can bind complement protein C1R, suppresses coagulation and fibrinolysis, and provides protection against IBD [78] and CRC [79]. These human proteins are also differentially targeted in healthy patients by the transcriptional regulator spo0A in Lactobacilli, Streptococci, and *F. prausnitzii* (Fig. 4A, Additional file 7: Table S6). Imatinib mesylate (brand name: Gleevec) targets several Src family tyrosine kinases, including LCK, which is involved in T cell development and has a recognized role in inflammation [80]. Bacterial proteins targeting these same kinases are consistently enriched in healthy controls across both IBD and three CRC cohorts we analyzed (Fig. 4B, Additional file 7: Table S6). In addition, imatinib can also halt the proliferation of colonic tumor cells and is involved generally in inflammatory pathways, through its inhibition of TNF-alpha production [81]. Consistent with the idea that bacterial proteins alone can act therapeutically, there is direct evidence for at least two commensal proteins which induce physiological effects on the host when delivered by oral gavage: purified *A. muciniphila* Amuc_1100 and *F. prausnitzii* MAM to ameliorate glucose intolerance and colitis, respectively [3, 62]. We suspect that this may extend to additional commensal proteins.

We also find instances where the off-label effects or side effects associated with the drug match our microbiome-driven human protein association. For instance, the antimalarial drug artinemol targets human proteins that were found to be differentially targeted by IBD cohorts' microbiomes (in Franzosa et al.): the RNA helicase DDX5, puromycin-sensitive aminopeptidase (NPEPPS), annexin A2 (ANXA2), and the splicing factor SFPQ (Fig.4C, Additional file 7: Table S6). Whereas artinemol and related analogs have been shown to be effective at preventing dextran sulfate-induced colitis in mice [82, 83] and wormwood, its natural source, has been established as a herbal treatment for IBD [84], microbiota-derived proteins have a greater association with IBD patients,

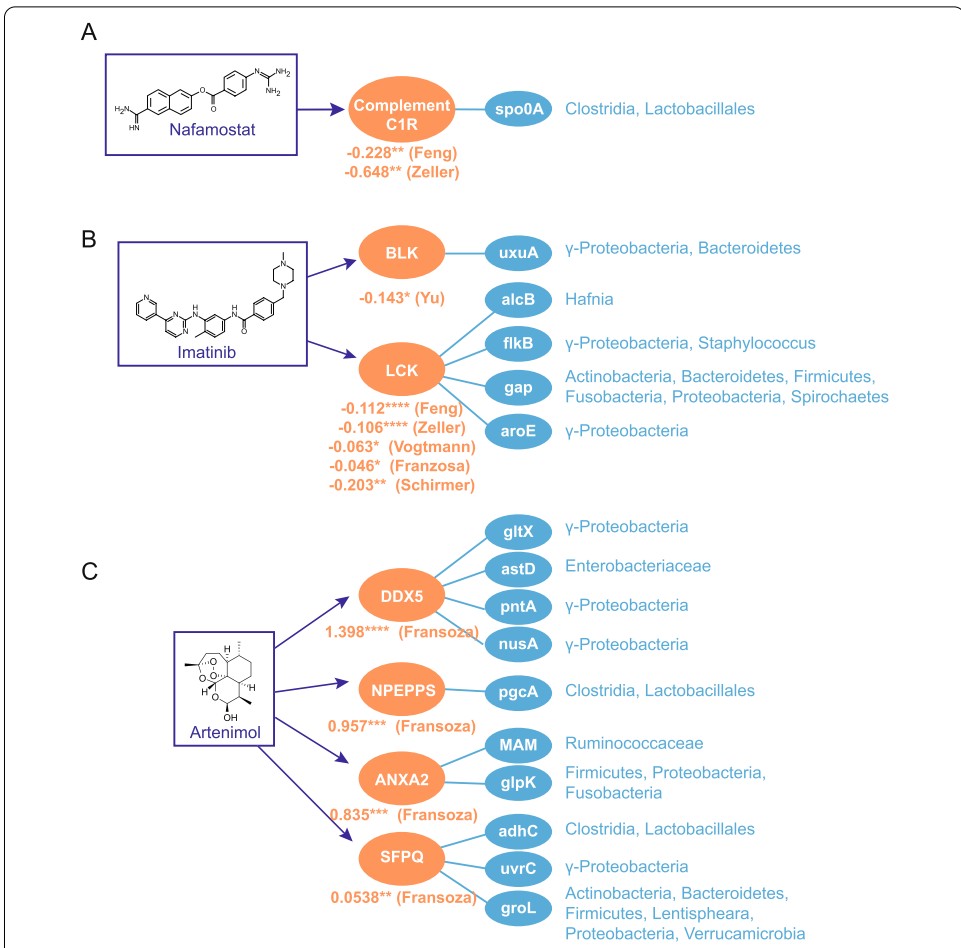

**Fig. 4** Human proteins targeted by gut commensal proteins include known therapeutic drug targets. **A** Nafamostat, **B** imatinib, and **C** artenimol target human proteins that are differentially targeted by bacterial proteins detected in the stated metagenomic studies. $\log_{10}$ relative mean summed abundances of bacterial interactors in patients versus controls are provided. p-values were calculated by the Mann-Whitney rank-sum test, * $p < 0.05$; ** $p < 0.01$; *** $p < 10^{-3}$; **** $p < 10^{-4}$). Full taxa and UniRef numbers for all bacterial proteins shown are provided in Additional file 7: Table S6

suggesting that artinemol and commensal proteins may be acting on the same targets in opposing ways. Whereas the notion of microbiome-derived metabolites acting as drugs is well-appreciated [85, 86], this work broadens the scope of microbiome-derived drugs to include protein products acting through PPI.

## Conclusion

This approach enables a high-throughput glimpse into the host-microbiome interactions, allowing for mechanistic inference and hypothesis generation from any metagenomic dataset. However, this network is far from complete. Estimates of a ratio of positive:negative host-pathogen PPIs are as high as 1:1000, or even 1:100 [87, 88]. Whether commensal microbiota interact with host proteins to anywhere near this extent will require substantial investigation. Few of the studies on which this interaction network is based were performed on commensal bacteria and intestinal tissue, and therefore, we may be missing interactions specific to our most intimately associated

bacteria. In addition to large-scale PPI studies involving commensal bacteria and their hosts, further in-depth studies will be needed to fully characterize these mechanisms, such as whether these bacterial proteins activate or inhibit their human protein interactors' pathways, and under what conditions these interactions take place. Pinpointing microbe-derived proteins like this that interact directly with human proteins will enable the discovery of novel diagnostics and therapeutics for microbiome-driven diseases, more nuanced definitions of the host-relevant functional differences between bacterial strains, and a deeper understanding of the co-evolution of humans and other organisms with their commensal microbiota.

## Methods

### Building a putative bacteria-human protein-protein interaction (PPI) network

Interactions were downloaded from the IntAct database [23], HPIdb 3.0 [25], and BioGRID [24] [June 2021] and supplemented with additional host-microbe interaction studies, whose interactions were added manually (PMIDs: 31227708, 34237247, 22213674, 18937849, 8900134, 17709412, 19047644, 23954158, 24335013, 24936355, 25680274, 26548613, 28281568, 29748286, 30072965, 30242281, 32566649, 32736072, 18808384, 22344444, 33820962, 31611645, 32051237, 18941224, 19627615, 3125250, 19752232, 21441512, 19542010, 11113124, 29335257, 21740499, 18541478, 9466265, 24204276, 23800426, 27302108, 25739981, 19907495, 31503404, 25118235, 25788290, 21699778, 26755725, 14625549). Only interactions with evidence codes that indicated binary, experimental determination of the interaction between UniProt identifiers with non-matching taxa were preserved, thereby excluding co-complex associations, small molecule interactions, and predicted interactions. Uniref100/90 clusters containing human proteins and Uniref50 cluster containing bacterial proteins were downloaded from UniProt [June 2021], to which interspecies protein interactors were mapped [89]. PPIs comprising one Uniref100/90 cluster containing human proteins and one Uniref50 cluster containing bacterial proteins were retained for downstream analyses. Within each UniRef50 bacterial cluster, we further filtered the sequences such that only bacterial members of the cluster within 70% sequence similarity to the experimentally verified protein were labeled as putative interactors. Sequence similarity was calculated using a Smith-Waterman local alignment with the BLOSUM62 matrix via python's parasail [90] library (v.1.1.17) and tallying the number of matches in the pairwise alignment that represent frequent substitutions (non-negative BLOSUM62 scores), divided by the length of the experimentally verified interactor.

### Processing of metagenomic shotgun sequencing data

The datasets used in this study, with the exception of the PRISM dataset [29], were curated as part of ExperimentHub [91] (Additional file 1: Table S1). Within each study, we removed samples that had abnormally low (less than $10^7$) reads. We downloaded all protein abundance matrices, annotated at the level of UniRef90 clusters via HUMAnN3 [92], and associated metadata. For PRISM, we processed data in a parallel manner, as outlined in Pasolli et al. [91]. For each study, we mapped UniRef90 bacterial clusters to UniRef50 clusters using DIAMOND [93] blastp, requiring greater than 90% sequence identity and greater than 90% coverage.

**Prioritization of disease-associated human targets**

For each patient, we identified bacterial proteins and aggregated their abundances according to their corresponding human protein interactors. In each study, we filtered out human proteins present in fewer than 5% of the cohort. To identify host-microbiome interactions that associate with disease, processed abundance matrices of putative human interactors were used to train a random forest machine learning classifier on the task of separating case and control patients and, after verifying that they achieve reasonable performance on the task using fivefold cross-validation with grid search-based hyperparameters tuning for each study (Additional file 2: Fig. S6), we extract the average feature importance from 100 iteratively trained class-balanced classifiers. Having used the scikit-learn [94] implementation of the random forest algorithm, feature importance corresponds to the average Gini impurity of the feature in all splits that it was involved in. Gini feature importance is a powerful prioritization tool, as it can capture the multivariate feature importance (whereas simple metrics like log-odds ratio and corrected chi-squared statistics only capture univariate feature importance). We created a disease-associated set for the proteins that had feature importances above the top 90th percentile. This cut-off was chosen as it was a conservative metric, which conserved model performance across studies (Additional file 2: Fig. S6C). Although model performance, specifically F1 score, varied by cohort, in all cases, model performances increase after we subset features to include only those with Gini importances over the 90th percentile.

We evaluated several alternative models for this study. Models performed on the bacterial protein abundances rather than the human protein abundances had marginally poorer performance (Additional file 2: Fig. S11) and were less interpretable. We compared the performance of RF with other machine learning models, namely logistic regression and support vector machines (SVM). These models can be sensitive to feature collinearity, so we reduced the multicollinearity by preclustering highly correlated features (Pearson's correlation coefficient > 0.8) in each metagenomic study and only included cluster representatives for training each model. We compared model performances with preclustered human protein abundances by model_selection.cross_validate in scikit-learn [94]. Hyperparameters of models were tuned using grid search to achieve their best performances. Random forest outperformed other machine learning models, including support vector machines (SVM) and logistic regression (Additional file 2: Fig. S12). Clustering of features did not drastically alter random forest model performance or feature importance rankings (Additional file 2: Fig. S13A). Most of the important features did not have high multicollinearity (Additional file 2: Fig. S13B) and were identified in the non-clustered models (Additional file 2: Fig. S13C-D). To improve interpretability, we chose to select the disease-associated set using RF trained with unclustered protein abundances. As an alternative to calculating human protein abundances by summing the total bacterial abundances of their interactors, we tested the effect of first normalizing bacterial abundances by their respective number of putative human interactors. This did not qualitatively change the conclusions drawn from our analyses.

**Human pathway annotation and enrichment analysis**

Disease annotations were extracted from all of GDAs from DisGeNET [95] (June 2021). We additionally downloaded all reviewed human proteins from Uniprot [96] (June 2021), annotating them in the same manner, in order to accurately compare background label

frequencies. Lacking a simple hierarchy of disease, we binned similar disease terms into the 5 larger categories relevant to our study. Human protein identifier labels are provided in Additional file 2: Supplementary Note 1. We performed pathway enrichment analysis using QIAGEN's Ingenuity® Pathway Analysis software (IPA®, QIAGEN Redwood City, CA, USA, www.qiagen.com/ingenuity). Sets of human proteins (HBNET, Detected, Disease-associated) were uploaded as UniProt identifiers into the desktop interface and submitted to their webserver for Core Enrichment Analysis was conducted only on human tissue and cell lines and IPA's stringent evidence filter. Pathways were considered enriched if they had Benjamini-Hochberg-corrected p values < 0.05. Subcellular locations for human proteins were obtained using GO Cellular Component terms associated with each protein in UniProt. We aggregated the following GO terms: Extracellular: Extracellular region (GO:0005576), Extracellular matrix (GO:0031012); Membrane: Cell surface (GO:0009986); Membrane (GO:0016020), Cell junction (GO:0030054); Cell projection (GO:0042995); and Intracellular: Cytoplasm (GO:0005737); Cell body (GO:0044297); Nucleoid (GO:0009295); Membrane-enclosed lumen (GO:0031974); Organelle (GO:0043226); Endomembrane system (GO:0012505); and Midbody (GO:0030496). Tissue-specific RNA expression enrichment was performed using DAVID bioinformatics resources [97]. Additionally, tissue-specific protein localization data was downloaded from Human Protein Atlas version 20.1 [98]. We retained those with 'enhanced', 'supported' and "approved" reliability. We additionally annotated all human proteins with any known drug targets from the DrugBank database [99] and DrugCentral (June 2021) [100].

### Bacterial pathway, secretion, and taxonomy annotation

For the purposes of annotation, we selected the representative bacterial sequence of each cluster. If there was no bacterial representative, we sorted sequences by their status in Uniprot (reviewed/unreviewed) and by their length and chose the top sequence. Bacterial taxonomy information is associated with each UniRef90 cluster by HUMANN3 [92]. We submitted all bacterial protein sequences to the KofamKOALA [101] KEGG orthology search resource to obtain orthology and pathway annotations. To obtain secretion information, we used several sources: we submitted our bacterial sequences to EffectiveDB [102] in order to obtain predictions for EffectiveT3 (type 3 secretion based on signal peptide) and T4SEpre (type 4 secretion based on amino acid composition at the C-terminus). We used the single default cutoffs for T4SEpre, and chose the "selective" (0.9999) cutoff for EffectiveT3. We obtained predictions for Sec and Tat pathway secretion using SignalP 5.0 [103] for Gram-positive and Gram-negative bacteria using default settings. Transmembrane proteins or signal peptides were predicted using TMHMM [104] (v.2.0c), with a threshold of 19 or more expected number of amino acids in transmembrane helices. Localization to the cell wall was predicted using PSORTb 3.0 [105] with default settings. We annotated secretion systems in species associated with each bacterial cluster by examining the core or minimal components of each secretion system, by searching their genomes using KEGG orthologous groups for each system using string cutoffs (identity > 40%; *e*-value < 0.00001; coverage > 80%): T3SS: sctR (K03226), sctS (K03227), sctT (K03228), sctU (K03229), and sctV (K03230); T4SS: virB4 (K03199) and virD4 (K03205); Sec: secY (K03076), secE (K03073), and secG (K03075); and Tat: tatA (K03116) and tatC (K03118). We defined genomes in which have all minimal components of each system as organisms with functional corresponding secretion systems.

### Structural data for these microbiome-human PPIs

We measured the extent to which structural interfaces could be used to infer microbiome-human protein-protein interaction by using DIAMOND [93] to query all amino acid sequences submitted to PDB (identity > 70%; coverage > 50%). In order to identify interface residues between each pair of chains in the cocrystal structures, we first use NACCESS (http://www.bioinf.manchester.ac.uk/naccess/) to calculate the solvent accessibility of each residue in each chain. Residues with $\geq 15\%$ solvent-accessible surface area are considered surface residues. We then calculate the change in accessible surface area for each residue when other chains in the same crystal structures are introduced. Residues which have a change in solvent accessible surface area above 1 $\text{Å}^2$ are determined to be interface residues. Cases in which human protein and bacterial proteins match their respective chains exclusively are in Additional file 8: Table S7. We highlight one example in which there are uniquely mapped chains, where 1p0s chains H and E match human coagulation factor X and bacterial Ecotin, respectively (Additional file 2: Fig. S14).

To assess the conservation of interface residues across bacterial members of the same UniRef cluster, we downloaded a list of all PDB structures which contain both human proteins and bacterial proteins, the UniRef50 cluster identifier for the bacterial protein, and all protein sequences in the corresponding cluster that also originate from bacterial proteomes from Uniprot. Using Clustal Omega, we then generated multiple sequence alignments for all the members of each UniRef50 cluster. We calculated interface residues on all pairs of chains in each structure and measured the BLOSUM62 similarity between bacterial interface residues and their corresponding amino acids in their respective UniRef50 cluster MSA. We then calculated the Jensen-Shannon divergence on the columns of the MSA containing interface residues.

## Supplementary Information

---

**Additional file 1: Table S1.** Extended information on known experimentally verified host-microbiome interactions with evidence for a role in cellular physiology and/or human health. Information on the interaction detection method for human-microbiome PPIs that have been shown to affect cell physiology and/or human health.

**Additional file 2..** This file contains Figures S1-S14.

**Additional file 3: Table S2.** Metagenomic samples used in this research. For each study, we list the sample numbers and labels in the cohort study.

**Additional file 4: Table S3.** Disease-associated human-microbiome PPIs. Human-microbiome PPIs are listed according to their UniProt and UniRef50 identifiers, human and bacterial protein names.

**Additional file 5: Table S4.** Number of human interactors according to the source of the experimentally-verified interactors. The number of human interactors, according to the species sourcing the initial experimentally verified interacting protein.

**Additional file 6: Table S5.** Human interactors that are known drug targets. For each disease-associated human protein, we list the drug interactor (annotated using DrugCentral and DrugBank) and the study in which it was found to be important.

**Additional file 7: Table S6.** Extended information for bacterial proteins targeting known drug targets in Fig. 4. Bacterial clusters depicted in Fig. 4 are listed with their UniRef number and detected taxa, according to HUMANN3.

**Additional file 8: Table S7.** Cocrystal structures representing interactions in our dataset. All pairs of detected bacterial proteins and human proteins in the nine metagenomic datasets that have BLASTp matches to two different chains within the same PDB cocrystal structure (totaling 8 bacterial protein clusters and 10 human proteins). This list includes structures with at least one chain exclusive to each bacterial and human proteins.

**Additional file 9: Table S8.** Experimental protein-protein interactions that were used for mapping to microbiomes.

**Additional file 10.** Review history.

### Acknowledgements
We wish to acknowledge members of the Brito lab, Indrayudh Ghosal, Giles Hooker, and Andy Clark for their thoughtful comments.

### Review history
The review history is available as Additional file 10.

### Peer review information

### Authors' contributions
H.Z., J.F.B, and I.L.B. conceptualized and designed the study and co-wrote the manuscript. The author read and approved the final manuscript.

### Funding
Ilana Brito is a Pew Scholar in Biomedical Sciences, a Packard Foundation Fellow, and a Sloan Foundation Research Fellow. Ilana Brito is funded by the National Institutes of Health (1DP2HL141007). This work was supported by the National Sciences Foudnation: Extreme Science and Engineering Discovery Environment (XSEDE) (BCS190008).

### Availability of data and materials
This work relies entirely on publicly accessible data. All metagenomic datasets are listed in Additional file 3: Table S2. All resources and databases used for annotation are listed in the Methods section. Disease-associated human-microbiome PPIs are listed in Additional file 4: Table S3. Scripts used for the metagenomic analysis are available at https://zenodo.org/record/6149203 [106] and https://github.com/britolab/Human_microbiome_PPI. All experimental PPIs were used for mapping to microbiomes are listed in Additional file 9: Table S8.

## Declarations

### Ethics approval and consent to participate
Not applicable.

### Consent for publication
Not applicable.

### Competing interests
The authors declare that they have no competing interests.

### Author details
[1]Department of Microbiology, Cornell University, Ithaca, NY, USA. [2]Meinig School of Biomedical Engineering, Cornell University, Ithaca, NY, USA.

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

## 