## [**Additional file 10.** Review history. · Genome Biology]

Review History

First round of review

Reviewer 1

Are you able to assess all statistics in the manuscript, including the appropriateness of statistical tests used? There are no statistics in the manuscript.

Comments to author:

Here the authors aim to uncover mechanisms of how microbes can cause disease through interactions between some of their proteins and those of the human host. They observe that the microbiome targets diverse human proteins involved in immune response, and other key functions. As the authors indeed note (and cite) earlier studies were carried out with a similar concept in mind. Here they explore proteins in the gut microbiome that have homology to interactors that bind human proteins. What sets this analysis apart from the earlier one is that because they exploit available datasets of already modeled protein-protein interactions, rather than predicting them based on structures as the earlier studies did, they were able to carry out a comprehensive analysis. Their analyses are broader and consequently enriched in interesting and new observations. As such, I support their publication. This type of analysis is important and can reveal one way through which the microbiome can cause disease. It provides specific interactions which experiments can then test.

Specifically, the authors collected human-microbe PPIs from multiple databases, defining "HBNet" set. From patient data, proteins homologous to the bacterial proteins are collected as "Detected" set. Then, Random Forest classifier was trained on patient data. The high-importance (by average Gini impurity) genes are collected as "Disease-associated" set. Downstream analyses are performed based on these three sets of genes (proteins). HBNet homologs were mapped to 821 species mainly in human gut microbiomes. The authors then tried to find enrichment in localization. Other than bone marrow, there was no significant localization enrichment found for the "Detected" and "Disease-associated" proteins. The authors suspect that the experimental design may be biased towards general expression patterns.

Additional minor comments:

The authors write that "Disease-associated human proteins contain significant proportion of secreted proteins, and many of them are exclusively intracellular. Using various computational tools, 12.2% and 16.6% "Disease-associated" microbial proteins are predicted to be secreted and transmembrane, respectively."

Initially, this sentence was confusing, whether it means 16.6% of "Disease-associated" microbial proteins or human proteins. As human proteins were discussed in the previous paragraph, I guess it is for microbial proteins. But then microbial transmembrane proteins are allowing for direct contact with live or intact bacteria? Does this mean bacteria-bacteria contact? I assume they meant bacteria-host cell contact here, but the sentence is written unclearly. First, the subject of the sentence is bacterial proteins, then they are "potentially allowing for direct contact with live or intact bacteria, ...," which suggests bacteria-bacteria contact.

At few other spots the sentences were confusing as well. For example, Lines 187-190: "Based on the bacterial cluster representatives from in the microbiomes from these nine cohorts, we 188 find evidence that at least 79.0% and 58.9% of disease-associated clusters predicted to be secreted by 189 T3SS and T4SS, respectively, have representative proteins found in organisms with the corresponding 190 secretion systems (T6SS were excluded due to the limited availability of prediction tools)." Although it is not the major contribution of this work, this sentence was confusing. At first sight, 79% and 58.9% seem very large proportions. Then, they are actually 79% of "associated" proteins that are predicted to contain signal peptides (secreted) and so on. It will be good to provide the actual counts here (e.g. how many proteins/clusters) as they did in previous paragraphs.

The authors note that "One of the major advantages of our work is that through this new interaction network, we vastly improve our ability to annotate host-relevant microbiome functions". I agree. It is one of the major contributions of this work: more functional annotations of microbial proteins. Notably, the predicted interactions are based only on sequence homology, and the gene prioritizations are based on a classical machine learning approach. Thus, the predicted microbe-disease associations are also based on predictions, that lack experimental evidence or structural details. This needs to be mentioned.

In Line 218 the authors note that "Microbiome proteins may act on human targets as therapeutic drugs" and that many of the observed disease-associated human proteins that are known drug targets. I expect that there will be many more such examples found in the predicted interactions. Since the predicted interactions are based on homology to known interactions, at least some of these could be elaborated further based on literature search. If the authors can also find more direct evidence it will be an even stronger contribution.

It will also be good to update the references. Please see doi: 10.1016/j.jmb.2020.01.025; doi: 10.1007/978-1-4939-8736-8_18

With a minor revision this will be a very nice paper that with the increasing community focus on the microbiome I hope will attract much attention.

Reviewer 2

Are you able to assess all statistics in the manuscript, including the appropriateness of statistical tests used? Yes, and I have assessed the statistics in my report.

Comments to author:

In this manuscript by Zhou et al, putative protein-protein interactions between humans and the gut microbiome are characterized in health and disease. Specifically, a database of inter-domain protein interactions was generated, microbial proteins from metagenomic sequencing data was mapped to that database, and the interacting human proteins were then used in a random forest model between healthy and affected individuals for several different diseases. The function,

location, and health effects of these human genes are explored and then mapped back to the bacterial partners. This paper uses a clever bioinformatic approach to an important open problem in microbiome studies, which is identifying molecular mechanisms that link microbiome perturbations to disease. It's a unique approach, and in many ways opens many more questions to be explored than it answers (which is a good thing!). While I am largely enthusiastic about this manuscript, there are a handful of areas that should be clarified or expanded for maximum utility.

Comments:

1. There are several aspects of the random forest approach that should be further clarified for the reader:
 - a. Page 19: What was the metric for "reasonable performance" of the RF classifiers? Some of the results from the supplementary figure look barely better than chance. These details should be outlined in the methods section.
 - b. Were the effects of multicollinearity assessed when performing the identification of disease-associated features in the Random Forest model? Correlated features in an RF are fine for classification purposes but can dilute each other's effects when trying to interpret the predictor variables. This is a potential issue that should be clarified in the methods, either by describing how collinearity isn't an issue or the steps taken to reduce any issues arising from it.
 - c. The choice of Random Forests for this application isn't particularly well motivated in the manuscript over simpler and more comprehensive methods. One tricky aspect of interpreting variables from RF or similar classification methods is this issue of missing potentially relevant features by only examining the ones with the most discriminatory power. Other microbe-human proteins may be interacting in disease that are not picked up in that top 90th percentile threshold that was chosen. Why is Random Forest chosen for this over something more straightforward like a logistic model? Power issues? Are results drastically different if disease associations are identified through other methods?
 - d. Given that RF was the chosen method, how much better at either discriminating or identifying putative human interaction proteins is the pipelines as described vs. if a RF of the bacterial proteins are used and then mapped to human proteins? Presumably there is better power to identify these human genes using the method in the paper, but some quantification of how much better could be an effective way to demonstrate the utility of this approach.
2. The fact that existing PPI databases are strongly biased towards pathogens raises questions about the PPIs that might be missed here from commensals. Using the percent identity match is a clever way to identify similar commensal proteins to those found in pathogens, but what about classes of proteins not found in pathogens? Could protein families that appear in the metagenomic datasets that appear unrepresented in the PPI databases be identified such that they could be prioritized for future study? Is there a way to estimate/extrapolate how the results in the study compare to what is feasible given the diversity of genes in the microbiome? For example, could the estimated number of genes in the gut microbiome be calculated and combined with the number of human genes to create an upper limit on how many interactions are possible? In many ways this manuscript opens more questions than it answers, and analysis explorations like these could serve as a jumping off point to justify further PPI/bacterial protein characterization.
3. Is the PPI database generated here publicly available? This could be a very useful public resource.

Minor comments:

1. p19: Line 650 - 651: This sentence is unclear as written, please reword.
2. p24: The caption for Figure S1 in the paper doesn't line up with Figure S1 (looks like a figure is missing?)
3. p26: The caption for Figure S10 doesn't line up with the submitted figure.

Response to Reviewer's Comments

Host-microbiome protein-protein interactions capture disease-relevant pathways

Hao Zhou, Juan Felipe Beltrán, Ilana Lauren Brito

Editorial comments:

Thank you very much for submitting your manuscript to Genome Biology, and please accept my apologies for the delay in replying to you about it. It has now been seen by two referees and their comments are accessible below.

As you will see, the reports are broadly favorable, so we are interested in publishing the manuscript, but we feel that the issues raised must be addressed in full, in the form of a revised manuscript, before we make a firm commitment to publication. When revising your manuscript, please ensure that all the points raised by both referees are addressed.

When revising the manuscript, please also ensure that the manuscript is formatted according to our instructions and that editable figures are provided with the revised manuscript. Please see <https://genomebiology.biomedcentral.com/submission-guidelines/preparing-your-manuscript/research> for further details. We also require all data to be deposited to a relevant repository prior to resubmission, although you may supply a reviewer access token if desired. Please note that if we decide to publish your manuscript we will require that all data be made publicly accessible and the 'live' accession numbers included in a separate Availability of Data and Materials section of the manuscript.

We thank the editor for the opportunity to revise our manuscript. We have responded to both reviewers' comments and revise the manuscript accordingly.

Reviewer reports:

Reviewer #1:

Here the authors aim to uncover mechanisms of how microbes can cause disease through interactions between some of their proteins and those of the human host. They observe that the microbiome targets diverse human proteins involved in immune response, and other key functions. As the authors indeed note (and cite) earlier studies were carried out with a similar concept in mind. Here they explore proteins in the gut microbiome that have homology to interactors that bind human proteins. What sets this analysis apart from the earlier one is that because they exploit available datasets of already modeled protein-protein interactions, rather than predicting them based on structures as the earlier studies did, they were able to carry out a comprehensive analysis. Their analyses are broader and consequently enriched in interesting and new observations. As such, I support their publication. This type of analysis is important and can reveal one way through which the microbiome can cause disease. It provides specific interactions which experiments can then test.

Specifically, the authors collected human-microbe PPIs from multiple databases, defining "HBNet" set. From patient data, proteins homologous to the bacterial proteins are collected as "Detected" set. Then, Random Forest classifier was trained on patient data. The high-importance (by average Gini impurity) genes are collected as "Disease-associated" set. Downstream analyses are performed based on these three sets of genes (proteins). HBNet homologs were mapped to 821 species mainly in human gut

microbiomes. The authors then tried to find enrichment in localization. Other than bone marrow, there was no significant localization enrichment found for the "Detected" and "Disease-associated" proteins. The authors suspect that the experimental design may be biased towards general expression patterns.

Thank you so much for recognizing the importance of our work. We have revised the manuscript according to your comments.

Additional minor comments:

1. The authors write that "Disease-associated human proteins contain significant proportion of secreted proteins, and many of them are exclusively intracellular. Using various computational tools, 12.2% and 16.6% "Disease-associated" microbial proteins are predicted to be secreted and transmembrane, respectively."

Initially, this sentence was confusing, whether it means 16.6% of "Disease-associated" microbial proteins or human proteins. As human proteins were discussed in the previous paragraph, I guess it is for microbial proteins. But then microbial transmembrane proteins are allowing for direct contact with live or intact bacteria? Does this mean bacteria-bacteria contact? I assume they meant bacteria-host cell contact here, but the sentence is written unclearly. First, the subject of the sentence is bacterial proteins, then they are "potentially allowing for direct contact with live or intact bacteria, ...," which suggests bacteria-bacteria contact.

We apologize for the confusion and we have made the following clarification: The 16.6% refers to microbial proteins.

Line 177-181: "... Another 16.6% of disease-associated microbiome proteins are predicted to be transmembrane, albeit with unknown orientation. Surface localization would potentially allow for direct contact between human proteins and either live or intact bacteria, or proteins on the surface of bacterially-produced membrane vesicles."

At few other spots the sentences were confusing as well. For example, Lines 187-190: "Based on the bacterial cluster representatives from in the microbiomes from these nine cohorts, we 188 find evidence that at least 79.0% and 58.9% of disease-associated clusters predicted to be secreted by 189 T3SS and T4SS, respectively, have representative proteins found in organisms with the corresponding 190 secretion systems (T6SS were excluded due to the limited availability of prediction tools)." Although it is not the major contribution of this work, this sentence was confusing. At first sight, 79% and 58.9% seem very large proportions. Then, they are actually 79% of "associated" proteins that are predicted to contain signal peptides (secreted) and so on. It will be good to provide the actual counts here (e.g. how many proteins/clusters) as they did in previous paragraphs.

We appreciate the reviewer's comments. We agree that actual counts will help readers digest the message here and we have modified the manuscript as below:

Line 188-191: "*Based on the bacterial cluster representatives from in the microbiomes from these nine cohorts, we find evidence that at least 79.0% (94/119) and 58.9% (20/34) of disease-associated clusters predicted to be secreted by T3SS and T4SS, respectively, have representative proteins found in organisms with the corresponding secretion systems (T6SS were excluded due to the limited availability of prediction tools).*"

The authors note that "One of the major advantages of our work is that through this new interaction network, we vastly improve our ability to annotate host-relevant microbiome functions". I agree. It is one of the major contributions of this work: more functional annotations of microbial proteins. Notably, the predicted interactions are based only on sequence homology, and the gene prioritizations are based on a classical machine learning approach. Thus, the predicted microbe-disease associations are also based on predictions, that lack experimental evidence or structural details. This needs to be mentioned.

We thank the reviewer for this comment. We agree that this should be made explicit in the manuscript and we have revised lines 217-219: *"Although this method generates provocative hypotheses that directly link human-associated microbiota to disease, functional experiments and structural evidence are greatly needed to confirm these specific host-microbiome PPIs."*

In Line 218 the authors note that "Microbiome proteins may act on human targets as therapeutic drugs" and that many of the observed disease-associated human proteins that are known drug targets. I expect that there will be many more such examples found in the predicted interactions. Since the predicted interactions are based on homology to known interactions, at least some of these could be elaborated further based on literature search. If the authors can also find more direct evidence it will be an even stronger contribution.

We agree that this is an exciting feature of the work. In the text, we discuss a number of human proteins targeted by both drugs and microbial proteins (lines 222-243): C1R, LCK and other Src family tyrosine kinases, DDX5, SFPQ, ANXA2 and NPEPPS. We provide the examples of Amuc_1100 and MAM, which show ameliorative benefits for glucose tolerance and inflammation, respectively, in mouse models. We searched the literature for those in whose mechanisms of action reflected the diseases in which we observed the specific association, and highlighted those that could be clearly explained. In addition, we provide a table (Table S5) with additional human proteins that are both targeted by gut microbiome proteins and serve as therapeutic drug targets.

It will also be good to update the references. Please see doi: 10.1016/j.jmb.2020.01.025; doi: 10.1007/978-1-4939-8736-8_18

Thank you for alerting us. We have updated the references accordingly.

With a minor revision this will be a very nice paper that with the increasing community focus on the microbiome I hope will attract much attention.

Thank you again for your thoughtful comments and interest in our work.

Reviewer #2: In this manuscript by Zhou et al, putative protein-protein interactions between humans and the gut microbiome are characterized in health and disease. Specifically, a database of inter-domain protein interactions was generated, microbial proteins from metagenomic sequencing data was mapped to that database, and the interacting human proteins were then used in a random forest model between healthy and affected individuals for several different diseases. The function, location, and health effects of these human genes are explored and then mapped back to the bacterial partners. This paper uses a clever bioinformatic approach to an important open problem in microbiome studies, which is identifying molecular mechanisms that link microbiome perturbations to disease. It's a unique approach, and in many ways opens many more questions to be explored than it answers (which is a good thing!). While I am largely enthusiastic about this manuscript, there are a handful of areas that should be clarified or expanded for maximum utility.

Thank you for your enthusiasm and taking the time to review our manuscript! We have addressed all your comments and revised our manuscript accordingly.

Comments:

1. There are several aspects of the random forest approach that should be further clarified for the reader:

a. Page 19: What was the metric for "reasonable performance" of the RF classifiers? Some of the results from the supplementary figure look barely better than chance. These details should be outlined in the methods section.

We agree that this could be clearer. The model performances vary across studies and two of the studies (Feng *et al.* and Qin *et al.*) had F1 scores at 0.54 and 0.56, respectively). In all cases, model performances increase after we subset the features to include only those with Gini importances over the 90th percentile. Despite variable performances, we achieve disease cohort-specific enrichment of proteins with associations concurrent with that disease (Figure S7), which we believe is strong evidence that our model is recouping disease-relevant signals. To acknowledge the variability in model performance across cohorts, we have added the text (lines 307-309): *"Although model performance, specifically F1 score, varied by cohort, in all cases, model performances increase after we subset features to include only those with Gini importances over the 90th percentile."*

b. Were the effects of multicollinearity assessed when performing the identification of disease-associated features in the Random Forest model? Correlated features in an RF are fine for classification purposes but can dilute each other's effects when trying to interpret the predictor variables. This is a potential issue that should be clarified in the methods, either by describing how collinearity isn't an issue or the steps taken to reduce any issues arising from it.

We thank the reviewer for these comments. To assess the effects of multicollinearity, we performed an experiment to compare the model performances and feature selection of our current models with those in which we pre-clustered highly correlated features (Pearson's correlation coefficient > 0.8) in each metagenomic study, including a single cluster representative in training. We found that the effect of multicollinearity did not affect our original results significantly (new Figure S13). Using the same feature importance cutoff (90th percentile), 93.6% (412/440) of important features were recaptured in the original model. We examined this further and found that important features generally belonged to clusters with few multicollinear features. We found that interpreting these multicollinear clusters across studies was challenging, as proteins may cluster differently across cohorts. We therefore chose to continue using the unclustered protein abundances in the models. We include this explanation in our discussion of alternative models, including SVM and logistic regression in lines 313-324.

c. The choice of Random Forests for this application isn't particularly well motivated in the manuscript over simpler and more comprehensive methods. One tricky aspect of interpreting variables from RF or similar classification methods is this issue of missing potentially relevant features by only examining the ones with the most discriminatory power. Other microbe-human proteins may be interacting in disease that are not picked up in that top 90th percentile threshold that was chosen. Why is Random Forest chosen for this over something more straightforward like a logistic model? Power issues? Are results drastically different if disease associations are identified through other methods?

Thank you for this comment. We chose the 90th percentile of Gini importances because it was a conservative estimate that conserved model performances in some cohorts. We have added the following text (lines 308-309): “*This cut-off was chosen as it was a conservative metric, which conserved model performance across studies (Fig. S6C).*” and we have added an image to Figure S6 (new Figure S6C) to illustrate model performance as a function of percentile cut-off.

To answer your question about the choice of RF over other types of modeling approaches, we now include model performances for a logistic regression and SVM (new Figure S12) for comparison. RF outperformed these models, as it is able to uncover nonlinear signals and predict high-dimensional tasks with a higher degree of efficiency and interpretability. We have also added commentary in lines 315-322: “*We compared the performance of RF with other machine learning models, namely logistic regression and support vector machines (SVM). These models can be sensitive to feature collinearity, so we reduced the multicollinearity by preclustering highly correlated features (Pearson’s correlation coefficient > 0.8) in each metagenomic study and only included cluster representatives for training each model. We compared model performances with preclustered human protein abundances by model_selection.cross_validate in scikit-learn (93). Hyperparameters of models were tuned using grid search to achieve their best performances. Random forest outperformed other machine learning models, including support vector machines (SVM) and logistic regression (Fig. S12).*”

d. Given that RF was the chosen method, how much better at either discriminating or identifying putative human interaction proteins is the pipelines as described vs. if a RF of the bacterial proteins are used and then mapped to human proteins? Presumably there is better power to identify these human genes using the method in the paper, but some quantification of how much better could be an effective way to demonstrate the utility of this approach.

Thanks for your comment. We have added a supplemental figure for the analysis using RF of the bacterial proteins (new Figure S11), which showed slightly lower performance than models using human proteins. We revised our manuscript in line 313-314: “*Models performed on the bacterial proteins abundances rather than the human protein abundances had marginally poorer performance (Fig. S11).*”

2. The fact that existing PPI databases are strongly biased towards pathogens raises questions about the PPIs that might be missed here from commensals. Using the percent identity match is a clever way to identify similar commensal proteins to those found in pathogens, but what about classes of proteins not found in pathogens? Could proteins families that appear in the metagenomic datasets that appear unrepresented in the PPI databases be identified such that they could be prioritized for future study? Is there a way to estimate/extrapolate how the results in the study compare to what is feasible given the diversity of genes in the microbiome? For example, could the estimated number of genes in the gut microbiome be calculated and combined with the number of human genes to create an upper limit on how many interactions are possible? In many ways this manuscript opens more questions than it answers, and analysis explorations like these could serve as a jumping off point to justify further PPI/bacterial protein characterization.

Thanks for posing these interesting questions. We acknowledge in the text that the network is far from complete. We agree that estimating the upper limit of PPIs between host and microbiome will help inform future work. Considering the roughly 4.5 million unique Uniref90 clusters found in the nine metagenomic studies we included and the 20,000-25,000 protein-coding genes within the human genome, the number of possible interactions is $\sim 10^{11}$. Other researchers have

suggested a ratio of positive:negative interactions is as high as 1:1000 or even 1:100 (Kshirsagar *et al.* 2013, 2015), one estimates that there could be as much as 10^8 - 10^9 non-redundant host-microbiome PPIs. We have added this to the text (lines 250-252): “*Estimates of a ratio of positive:negative host-pathogen PPIs are as high as 1:1000, or even 1:100. Whether commensal microbiota interact with host proteins to anywhere near this extent will require substantial investigation.*”

- Kshirsagar, Meghana, Jaime Carbonell, and Judith Klein-Seetharaman. "Multitask learning for host-pathogen protein interactions." *Bioinformatics* 29.13 (2013): i217-i226.
- Kshirsagar, Meghana, et al. "Techniques for transferring host-pathogen protein interactions knowledge to new tasks." *Frontiers in microbiology* 6 (2015): 36.

3. Is the PPI database generated here publicly available? This could be a very useful public resource.

We're glad you think so! All of the datasets we used are publicly available, in addition to the papers we manually curated. To improve the utility of this resource, we now include a table of binary protein-protein interactions that we used for mapping to microbiomes (new Table S8).

Minor comments:

1. p19: Line 650 - 651: This sentence is unclear as written, please reword.

Thank you for helping us improve the readability of our methods. This now reads: “*For each patient, we identified bacterial proteins and aggregated their abundances according to their corresponding human protein interactors.*” (lines 296-297)

2. p24: The caption for Figure S1 in the paper doesn't line up with Figure S1 (looks like a figure is missing?)

3. p26: The caption for Figure S10 doesn't line up with the submitted figure.

Thank you for calling this to our attention. We have addressed the misalignment of both figure captions.

Second round of review

Reviewer 2

The additions to this manuscript fully address the concerns I raised in my initial review. Adding evidence that RF is the most effective choice, adding nuance to the discussion, and the analysis of collinearity all strengthen the manuscript. I remain very enthusiastic about this paper and I suspect it will be the inspiration of many follow-up studies.

The one small comment I have would be to double check the AUROC values reported in Figure S12. They completely match between the SVM and the logistic regression. It could happen, but seems more likely it's a figure generation issue